# Spatiotemporal optical vortices with controllable radial and azimuthal quantum numbers

Xin Liu [1,2,8], Qian Cao [3,4,8], Nianjia Zhang[3], Andy Chong[5,6], Yangjian Cai [1,2] ✉ & Qiwen Zhan [3,4,7] ✉

Optical spatiotemporal vortices with transverse photon orbital angular momentum (OAM) have recently become a focal point of research. In this work we theoretically and experimentally investigate optical spatiotemporal vortices with radial and azimuthal quantum numbers, known as spatiotemporal Laguerre-Gaussian (STLG) wavepackets. These 3D wavepackets exhibit phase singularities and cylinder-shaped edge dislocations, resulting in a multi-ring topology in its spatiotemporal profile. Unlike conventional ST optical vortices, STLG wavepackets with non-zero $p$ and $l$ values carry a composite transverse OAM consisting of two directionally opposite components. We further demonstrate mode conversion between an STLG wavepacket and an ST Hermite-Gaussian (STHG) wavepacket through the application of strong spatiotemporal astigmatism. The converted STHG wavepacket is de-coupled in intensity in space-time domain that can be utilized to implement the efficient and accurate recognition of ultrafast STLG wavepackets carried various $p$ and $l$. This study may offer new insights into high-dimensional quantum information, photonic topology, and nonlinear optics, while promising potential applications in other wave phenomena such as acoustics and electron waves.

Light fields with twist wavefront of the form $e^{-il\theta}$, referred to as optical vortices (OVs), show a symmetric hollow intensity profile caused by the on-axis phase singularities (see Fig. 1a). The integer number $l$ is an azimuthal quantum number called topological charge. Allen et al. in 1992 discovered that spatial OVs, such as Laguerre-Gaussian beams, carry a longitudinal orbital angular momentum (OAM) proportional to the topological charge $l$. This OAM is parallel to the wavevector and propagation direction of the beam[1,2]. Since then, spatial OVs have been extensively studied and play a crucial role in various fields of physics, including light-matter interaction to manipulate the angular momentum of particles and control their motion[3,4], quantum optics to enable

entanglement and manipulation of photons for high-dimensional quantum information processing[5–7], telecommunications to offer increased channel capacity and improved immunity to signal degradation[8,9] and holographic data storage to elevate security and depth perception within holography[10,11], etc. Benefiting from recent successful developments in ultrafast optics, self-torque beams with time-varying longitudinal OAM have also been proposed through introducing spectrum related azimuthal indices into a pulsed beam[12–14].

Recent years have witnessed a breakthrough in the realm of OVs. Beyond the established longitudinal OAM parallel to the wavevector,

[1]Shandong Provincial Engineering and Technical Center of Light Manipulations and Shandong Provincial Key Laboratory of Optics and Photonic Device, School of Physics and Electronics, Shandong Normal University, Jinan, China. [2]Collaborative Innovation Center of Light Manipulations and Applications, Shandong Normal University, Jinan, China. [3]School of Optical-Electrical and Computer Engineering, University of Shanghai for Science and Technology, Shanghai, China. [4]Zhangjiang Laboratory, Shanghai, China. [5]Department of Physics, Pusan National University, Busan, Republic of Korea. [6]Institute for Future Earth, Pusan National University, Busan, Republic of Korea. [7]Westlake Institute for Optoelectronics, Fuyang, Hangzhou, China. [8]These authors contributed equally: Xin Liu, Qian Cao. ✉e-mail: yangjiancai@sdnu.edu.cn; qwzhan@usst.edu.cn

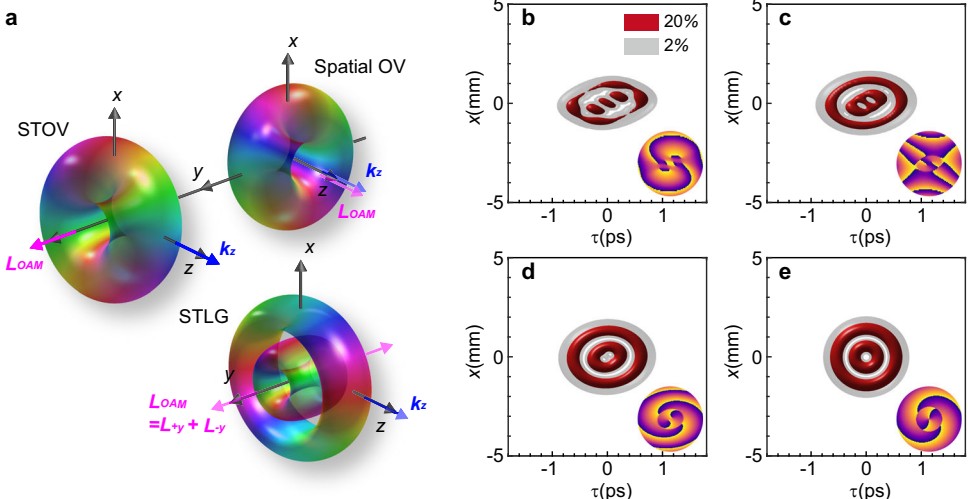

**Fig. 1 | Spatiotemporal Laguerre-Gaussian wavepackets. a** Schematic of a spatial optical vortex, spatiotemporal optical vortex and spatiotemporal Laguerre-Gaussian wavepacket. The OAM $L_{OAM}$ (pink arrows) in a spatial optical vortex is parallel to the wavevector $k_z$ (blue arrows), and thus named longitudinal OAM. The OAM in the STOV and STLG wavepacket are perpendicular to $k_z$ and thus called transverse OAM, while the transverse OAM in the later is composed of two kinds of direction-opposite components and thus named composite transverse OAM.

**b–e** 3D iso-intensity profiles of an STLG wavepacket of $p = 1$ and $l = +2$ and corresponding sliced phase patterns (at y = 0) at different locations $z = 200$, 400, 600 and 800 mm. The parameters are set as $\lambda_0 = 1.03$um, $w_1 = 0.4$ mm, $\gamma = 50$, $L = 800$ mm, the corresponding spectrum width is 8 rad/ps and positive GDD is $\alpha = 9,500$fs$^2$. The STLG of $p = 1$ and $l = +2$ **e** is faithfully synthesized at $L = 800$ mm and has a spatial width of 0.8 mm and temporal width of 182 fs.

the existence of transverse OAM perpendicular to the propagation direction has been theoretically and experimentally confirmed in a spatiotemporal optical vortex (STOV)[15–18] (see STOV in Fig. 1a). Unlike spatial OVs, which exhibit on-axis phase singularities, STOVs manifest these singularities in space-time (Fig. 1a). Their successful generation through a spiral phase in the spatial frequency-frequency domain of pulsed beams opens exciting avenues for various fields[19,20]. This revolutionary progress propels research towards photonic topology that explores topological phenomena in light by exploiting ST-coupled wavepackets[21–24], structured ultrafast wavepackets by tailoring the spatiotemporal properties of light pulses for precise control[25–30], nonlinear optics that harnesses the unique characteristics of STOVs for enhanced nonlinear interactions[31–33], nanophotonics that integrates STOVs with miniaturized optical devices for novel functionalities[34–36] and other wave phenomena by extending the principles of STOVs to other wave systems, such as acoustics and electrons[37–39]. However, the majority of current efforts have primarily focused on STOVs based on Gaussian-type wavepackets (see a recent review in ref. [40]).

Arguably the next most widely used spatial basis after Fourier are Laguerre-Gaussian (LG) and Hermite-Gaussian (HG) modes. Both define complete and orthonormal bases for light's spatial structure, offering an unbounded set of functions to represent any light field. LG modes, in particular, are characterized by two degrees of freedom: an azimuthal mode index ($l$) and a radial mode index ($p$). The azimuthal number is intricately linked to the well-studied intrinsic longitudinal OAM of photons[1]. The radial number ($p$), often referred to as the "forgotten quantum number", governs the beam's hyperbolic momentum[41,42]. Compared to other types of OVs, the ability to control both indices in LG-OVs has proven highly advantageous in fundamental physics and photonic engineering[43–48], particularly in classical and quantum information applications[7,49–52]. However, the space-time coupled LG mode with both radial and azimuthal quantum numbers remains relatively unexplored, presenting a promising frontier for further investigation.

In this work, we report on experimental synthesis of spatiotemporal Laguerre-Gaussian (STLG) wavepackets with well-defined radial and azimuthal quantum numbers through the integration of a 2D pulse shaper with a complex-amplitude modulation technique. Our

experimentally generated high-purity 3D STLG wavepackets exhibit multi-layered doughnut-shaped topological structures within their spatiotemporal profile (Fig. 1a). These remarkable STLG wavepackets possess two key features: 1. Screw singularities and edge dislocations: These unique spatial-temporal features endow the wavepacket with an intrinsic transverse OAM. 2. Composite transverse OAM: Theoretical analysis reveals that this transverse OAM comprises two directionally opposite components, arising from the multi-ring topological texture of the STLG wavepacket (see Fig. 1a). Furthermore, we demonstrate the conversion of STLG wavepackets into spatiotemporal Hermite-Gaussian (STHG) wavepackets by introducing precisely controlled spatiotemporal astigmatism, which opens up exciting new possibilities for manipulating and utilizing these novel light waveforms.

## Theoretical analysis and numerical simulations

The electric field of a generic pulsed beam is expressed as the product of the carrier at the central frequency and the slowly varying envelop (wavepacket), i.e. $E(t, x, y, z) = \Psi(t, x, y, z) \exp(i\omega_0 t - ik_0 z)$, in which $\omega_0 = k_0 c = 2\pi c/\lambda_0$ is the central angular frequency of the pulse. Under a scalar, paraxial and narrow bandwidth approximation, this wavepacket evolves in a uniform dispersive medium as[21]

$$\beta_2 \frac{\partial^2 \psi}{\partial \tau^2} - \frac{1}{k_0} \left( \frac{\partial^2 \psi}{\partial x^2} + \frac{\partial^2 \psi}{\partial y^2} \right) - 2i \frac{\partial \psi}{\partial z} = 0 \qquad (1)$$

where $\tau = t - z/v_g$ is the local time frame, $z$ is the propagation distance and $v_g$ is the group velocity. $\beta_2$ is the group velocity dispersion (GVD) coefficient of the dispersive medium. In the case of anomalous GVD medium with $\beta_2 = -k_0^{-1}$, a closed-form wavepacket solution of Eq. (1) is a spatiotemporal Laguerre-Gaussian wavepacket $\propto LG_p^l(\tau, x, y, z)$ [see Supplementary Section 1 for details].

To generate such STLG wavepackets, we introduce a spatial-spectral coupled polychrome Laguerre-Gaussian beam, which is given by

$$\Psi(\Omega, \xi) \simeq A_0 \left( \frac{\sqrt{2}r}{w_1} \right)^l \exp\left( -\frac{r^2}{w_1^2} \right) L_p^{|l|} \left( \frac{2r^2}{w_1^2} \right) \exp(-il\varphi) \qquad (2)$$

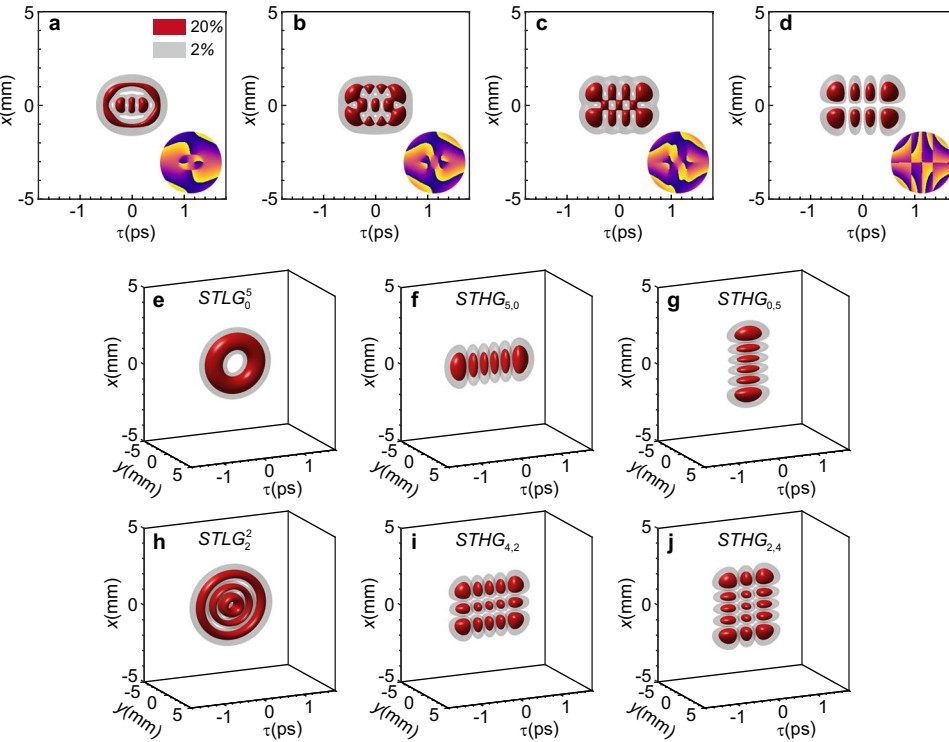

**Fig. 2 | Simulated mode conversion of the STLG wavepacket to STHG wavepacket by applying a spatiotemporal astigmatism. a–d** 3D iso-intensity profiles of the ST wavepacket (at $L = 800$ mm) and corresponding sliced phase patterns (at $y = 0$) under the different spatiotemporal astigmatism strength of $0.25\mu_0$, $0.5\mu_0$, $0.75\mu_0$ and $\mu_0$. $\mu_0 = 0.625$fs/(rad · mm) in this case, other parameters are set as the same as those in Fig. 1. The STHG wavepacket is faithfully synthesized at $\mu_0$ and has a spatial width of 3.5 mm and temporal width of 177 fs. **e**, An STLG wavepacket of $p = 0$ and $l = +5$ without spatiotemporal astigmatism. **f, g** The STLG wavepackets are mapping to the STHG wavepackets with a spatiotemporal astigmatism for an inverse azimuthal index $l = +5$ **f** and $l = -5$ **g**. **h** An STLG wavepacket of $p = 2$ and $l = +2$ without spatiotemporal astigmatism. **i, j** The STLG wavepackets are mapping to the STHG wavepackets with a spatiotemporal astigmatism for an inverse azimuthal index $l = +2$ **i** and $l = -2$ **j**.

where $r = \sqrt{\gamma^2\Omega^2 + \xi^2}$, $\varphi = \tan^{-1}(\xi/\gamma\Omega)$ and $\Omega = \omega - \omega_0$ is the reduced angular frequency; $w_1$ is the waist radius on the space-frequency plane; $L_p^{|l|}(\cdot)$ is the associated Laguerre polynomials of the order $p$ and $l$. $A_0$ is a complex constant and $\gamma$ is a scaling factor for controlling the balance between spatial diffraction and temporal dispersion during propagation. With the help of spatially diffractive and temporally dispersive propagation [see Supplementary Section 2], an STLG wavepacket is synthesized from the LG seed beam of Eq. 2, which has the following expression

$$\Psi_{STLG}(\rho, \theta) = B_0 \left(\frac{\sqrt{2}\rho}{w_2}\right)^l \exp\left(-\frac{\rho^2}{w_2^2}\right) L_p^{|l|}\left(\frac{2\rho^2}{w_2^2}\right) \exp(-il\theta) \quad (3)$$

where $B_0$ is a complex function. $\rho = \sqrt{\tau^2 + \alpha^2 x^2}$, $\theta = \tan^{-1}(\alpha x/\tau)$, $w_2 = 2\sqrt{w_1^{-2} + \alpha^2 w_1^2}$ and $\alpha = \gamma^2 k_0^2/2L$ is related to the diffractive and dispersive phase. The STLG wavepacket is characterized by two degrees of freedom $(p, l)$ and exhibits an intrinsic transverse OAM scaled by its azimuthal indices. The significant difference from a STOV is that the total transverse OAM of an STLG wavepacket is a vectorial superposition of direction-opposite transverse OAM (Fig. 1a). Such composite transverse OAM density distribution is caused by the radial quantum number of the Laguerre polynomial. The detailed theory is discussed in the Supplementary Section 3.

Equations (2) and (3) suggest that an STLG wavepacket can be synthesized by employing a spatial-spectral coupled polychrome LG seed beam. Figure 1b–e shows the numerical simulation results for synthesizing an STLG wavepacket at different diffraction distances in free space. In the simulation, the spatial-spectral LG beam of $p = 1$ and

$l = +2$ has a spatial width of 0.8 mm and a central wavelength of 1.03m with a spectrum width of 8 rad/ps and carries a positive group delay dispersion (GDD) $\alpha = 9500$fs². An STLG wavepacket (Fig. 1e) with $p = 1$ and $l = +2$ is faithfully generated at a location $L = 800$ mm, with a central maximum vortex ($l = +2$) core surrounded by $p$ vortex rings and each neighboring ring has a phase difference $\pi$, as shown in Fig. 1e. On the other hand, the STLG wavepacket is an exact solution of the paraxial wave equation. As such, the generated STLG wavepacket has a self-similar property when it evolves in a diffraction-dispersion matched medium [see Supplementary Section 4].

Inspired by the fact that LG basis can be easily transformed to HG through astigmatic optical components in the spatial domain[1,47], we can convert the STLG wavepackets to the STHG wavepackets by applying a spatiotemporal astigmatism [see Supplementary Section 5 for detailed theory]. Figure 2a–d shows the simulated mode conversion process from an STLG wavepacket to an STHG wavepacket under different astigmatic strengths. Unlike the cases without spatiotemporal astigmatism, the STLG wavepacket is firstly deformed and the singularities are dislocated in both time and space. This phenomenon is caused by the mismatch between temporal dispersion and spatial diffraction introduced by the spatiotemporal astigmatism. With appropriate spatiotemporal astigmatism, STLG wavepacket can be fully mapped into STHG wavepacket. The converted STHG wavepacket has $p + l$ order in time and $p$ order in space, respectively. Each neighboring lobe has a phase difference $\pi$, as shown in Fig 2d. In Fig. 2e–j, we show the simulated mode conversion of STLG wavepackets with different radial indices $p$ and azimuthal indices $l$ to STHG wavepackets. The STLG wavepackets with positive (or negative) azimuthal index are transformed to the STHG wavepackets of $p+l$ (or p) order in time and $p$ (or p-l) order in space and the results agree with the theoretical predictions very well.

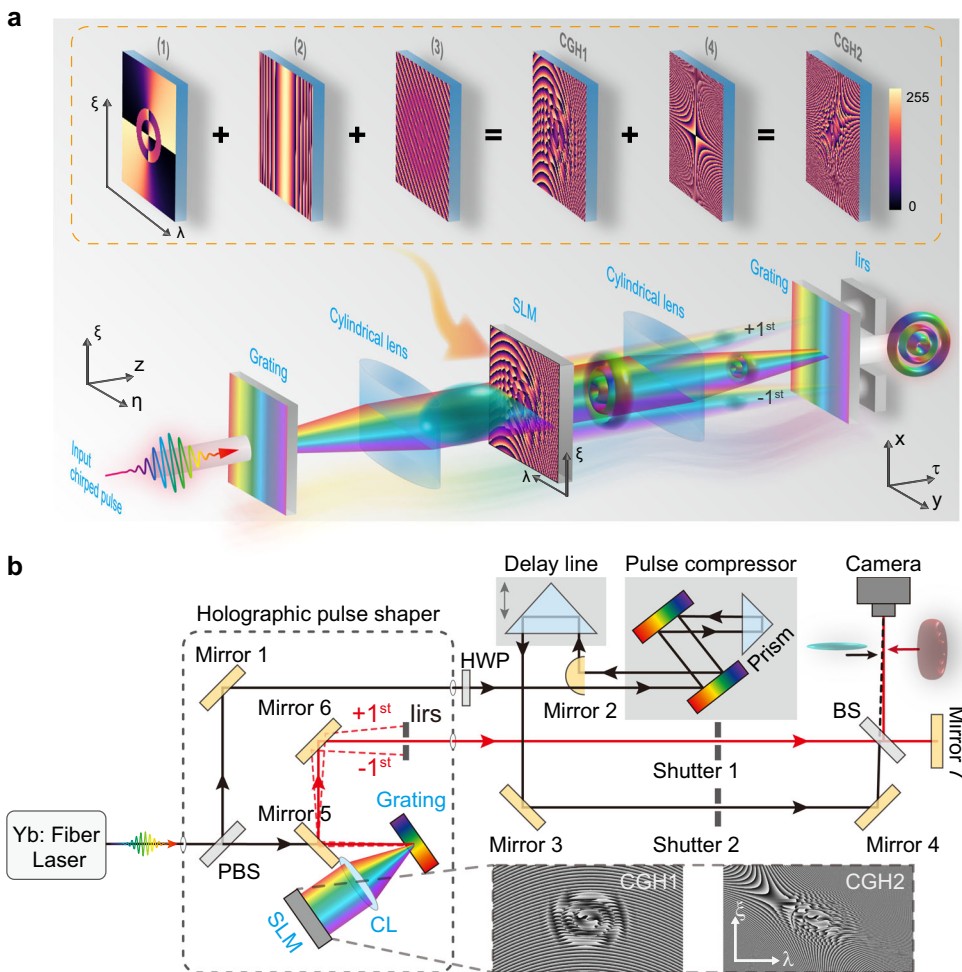

**Fig. 3 | Synthesis and characterization of STLG wavepackets and its mode conversion. a** Concept of STLG wavepacket synthesis based on complex-amplitude modulation technique. The computer-generated hologram (CGH) embedded on the SLM comprises four components: (1) the phase distribution of a spatial-spectral LG mode; (2) a GDD phase for managing the pre-chirp of input pulse and controlling the balance diffraction and dispersion; (3) a phase-only diffraction grating whose phase depth modulated by the modulus of Eq. (2) (see Methods); (4) an astigmatic phase. The synthetic CGH can realize a complex-amplitude modulation, wherein the grating phase depth and delay controls the intensity and wavefront of incident pulsed beam, respectively[57,58]. **b** Optical setup for synthesizing and characterizing STLG wavepackets. The apparatus comprises three sections: (i) 2D ultrafast holographic pulse shaper consisting of a diffraction grating, a cylindrical lens and an SLM; (ii) a pulse compressor system consisting of a parallel grating pair; (iii) a time delay line system for fully reconstructing 3D profile of generated ST wavepacket. PBS, polarized beam splitter; HWP, half-wave plate; BS, beam splitter.

## Experimental results

We experimentally synthesize the STLG wavepackets with the model described above. The basic concept (illustrated in Fig. 2a) combines 2D spatial light modulation and ultrafast pulse shaping[19,21,27]. The spectrum of a chirped input pulse beam is spatially spread along $y$-axis by a grating and collimates onto a spatial light modulator (SLM) through a cylindrical lens, so that each wavelength $\lambda$ is assigned to a position $y(\lambda)$ forming a 2D spatial-spectral domain. By applying a meticulously crafted hologram for the on-demand mode (see Methods), the spatial-spectral distribution is efficiently modulated at a short distance after the SLM. In this scheme, the desired ST wavepacket, as denoted by Eq. (2), is produced by selecting the zeroth order diffraction to circumvent spatial dispersion caused by diffractive deflection angles (see Methods and Supplementary Section 6). Recombining the spectrum via a second cylindrical lens and grating reconstitutes the pulse, the STLG wavepacket is realized after free-space diffraction of a distance $L$, where the wavepacket reaches a balance between the spatial diffraction and temporal dispersion leading to an ST-symmetry flying donut. If the STLG wavepacket propagates freely in space without a matched GVD, it will gradually degenerate from a donut shape into diagonally separated lobes, akin to the phenomenon observed in ref. 20.

The experimental apparatus is depicted schematically in Fig. 3b. A lab-built, mode-locked fiber laser emits a pulsed beam which has a spectral bandwidth of ~ 20 nm centered at a wavelength of 1030 nm and carries a linear frequency chirp [see Supplementary Section 7 for pulse characterization]. The light from the source is split into a probe pulse and an object pulse. The probe pulse passes through a pulse compressor consisting of a pair of parallel gratings and a right-angle prism, and is then de-chirped to a Fourier-transform-limited (FTL) pulse with a pulse width of ~122 fs [see Supplementary Section 7]. The object pulse goes into a folded 2D ultrafast pulse shaper that consists of a reflective grating (1200 lines mm⁻¹), a cylindrical lens (10 cm focus length on $y$ axis) and a phase-only reflective SLM (Holoeye GAEA-2, 3840×2160 pixels with a pitch of 3.74μm). The programmable SLM is situated in the spatial-spectral plane and imprints an elaborate phase pattern, illustrated in the bottom right corner (CGH1 and CGH2), which encodes the complex field amplitude of the polychrome beam of Eq. (2) (see Methods). The modulated beam is reflected back and the pulse is reconstituted by the grating to produce the STLG wavepacket on a specific plane where a CMOS camera is placed (around 1.2 m behind the pulse shaper in our setup). The undesired diffractive orders after the grating are blocked by an iris and only the zeroth order is selected and detected by the camera.

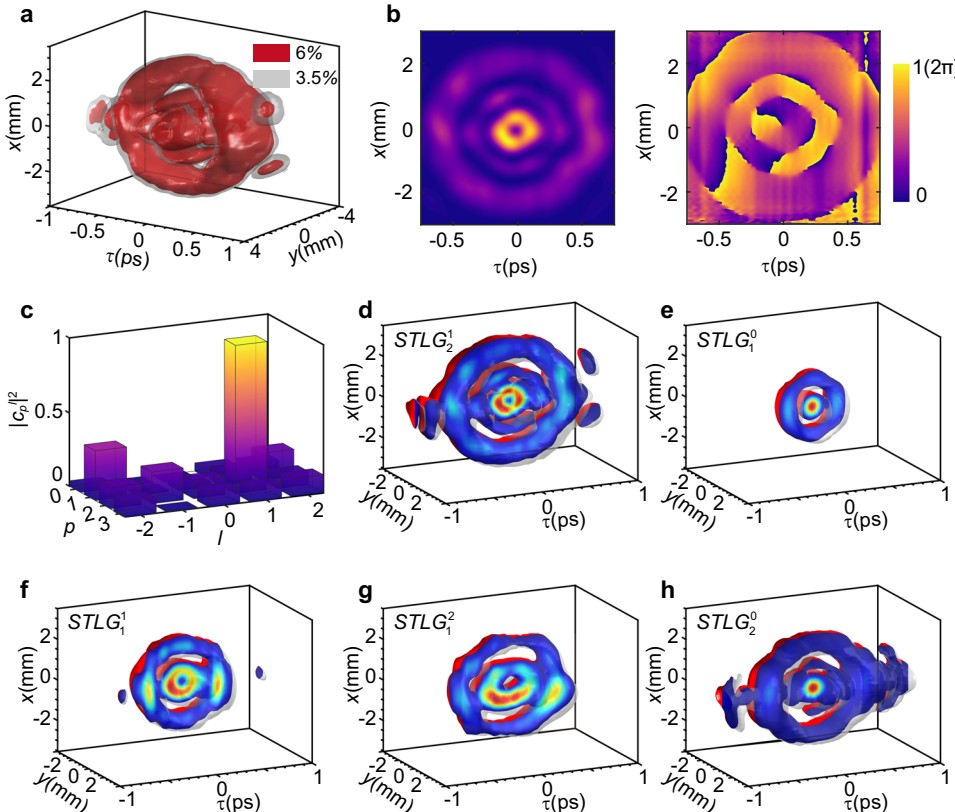

**Fig. 4 | Experimentally synthesized STLG wavepackets. a** Experimentally reconstructed 3D iso-intensity surface of an STLG wavepacket with radial index $p = 2$ and azimuthal index $l = +1$. The maximum intensity is normalized to 1. **b** Measured accumulated spatiotemporal intensity $\int |\Psi_{STLG}(\tau, x, y)|^2 dy$ (left) and sliced phase at $y = 0$ (right) distributions. The fitting spatial and temporal width are 1.4 mm and 300 fs respectively. **c** Modal weight analysis for the synthesized STLG wavepacket of $p = 2$ and $l = +1$. **d–h** Experimentally reconstructed 3D iso-intensity cross-section profiles of the STLG wavepackets with different modal indices $p$ and $l$. **d** $p = 2, l = +1$; **e** $p = 1, l = 0$; **f** $p = 1, l = +1$; **g** $p = 1, l = +2$ and **h** $p = 2, l = 0$. The isovalue is set to 0.06.

The ST wavepacket is synthesized in the far field via a time-delayed spatial propagation after the grating. Actually, we can readily realize a near field ST-wavepacket engineering by reducing the diffraction and dispersion phases. We use a 3D diagnostic measurement technique[53] to fully characterize the complex field information of the synthesized ST-wavepacket. Briefly, the object pulse is combined with the FTL probe pulse at a tilted angle to produce interference fringes pattern. These fringes record the spatial complex field of the generated STLG wavepacket at a certain time slice [see Methods and Supplementary Section 8 for detail]. By scanning the time delay line of the probe pulse, the spatiotemporal profile of the STLG wavepacket can be reconstructed from the delay-dependent fringes[19]. A representative reconstruction process is provided in the Supplementary Movie 1, in which the number of time slices is 100 frames with a time step of ~33 fs.

The experimental measurement and analysis of the synthesized STLG wavepacket of $p = 2$ and $l = +1$ are shown in Fig. 4a–c. Figure 4a presents the reconstructed spatiotemporal 3D iso-intensity surface, indicating a complicated spatiotemporally coupled 3D wavepacket, which resembles a centered wind vortex and a multi-layered doughnut-shaped topological structure. Figure 4b shows the accumulated spatiotemporal intensity $\int |\Psi(\tau, x, y)|^2 dy$ (left) and sliced phase at $y = 0$ (right) distributions (see Methods). These results clearly confirm the generation of an STLG wavepacket with a spiral phase ($l = +1$) and two outer rings ($p = 2$) with $\pi$ phase difference in both space and time. The purity of the synthesized wavepacket is analyzed by exploiting the orthonormality of LG modes [see Supplementary Section 9]. Figure 4c shows the normalized modal weight coefficients on radial and azimuthal indices. ~35.7% of the total power is located at the expected radial index ($p = 2$) and azimuthal index ($l = +1$). We also display the experimentally reconstructed

3D iso-intensity cross-section profile of the STLG wavepackets with different $p$ and $l$ in Fig. 4h, i for $p = 2$ and $l = +1$, $p = 1$ and $l = +0$, $p = 1$ and $l = +1$, $p = 1$ and $l = +2$, $p = 2$ and $l = +0$, respectively. The corresponding measured weight coefficients and accumulated intensity and phase distributions are provided in the Supplementary Sections 9 and 10. It is worth mentioning that such structured ST wavepackets are synthesized in the far-field via a time-delayed spatial propagation after the grating, which inevitably leads to a spatial/temporal broadening. The spatial and temporal width of the STLG wavepackets are determined by the time-dispersion and space-diffraction phases (see equation (21) in the Supplementary Section 2). Hence, we can regulate their spatial and temporal widths by manipulating the diffraction and dispersion phases on the SLM (Supplementary Section 11).

In addition, an STLG wavepacket can be converted into an STHG wavepacket by applying a strong spatiotemporal astigmatism. Figure 5a shows the experimentally reconstructed 3D iso-intensity surface of an STHG wavepacket that is converted from an STLG wavepacket with $p = 0$ and $l = +5$. The accumulated spatiotemporal intensity $\int |\Psi_{STHG}(\tau, x, y)|^2 dy$ (left) and sliced phase at $y = 0$ (right) distributions (see Methods) in space-time plane are plotted in Fig. 5b. Our experimental results show a Hermite-Gaussian type spatio-temporal intensity distribution which has $l + 1 = 6$ lobes in time with $\pi$ phase difference and ~250 fs temporal pitch for each other. We can flexibly generate STHG wavepackets with customized spatiotemporal indices based on the conversion rule described by Eq. (39) in the Supplementary Section 5, as shown in Fig. 5c–f. The corresponding accumulated intensity and phase distributions are presented in the Supplementary Section 12. In Supplementary Section 13, we record the mode conversion process for an STLG wavepacket of $p = 1$ and $l = +2$ to

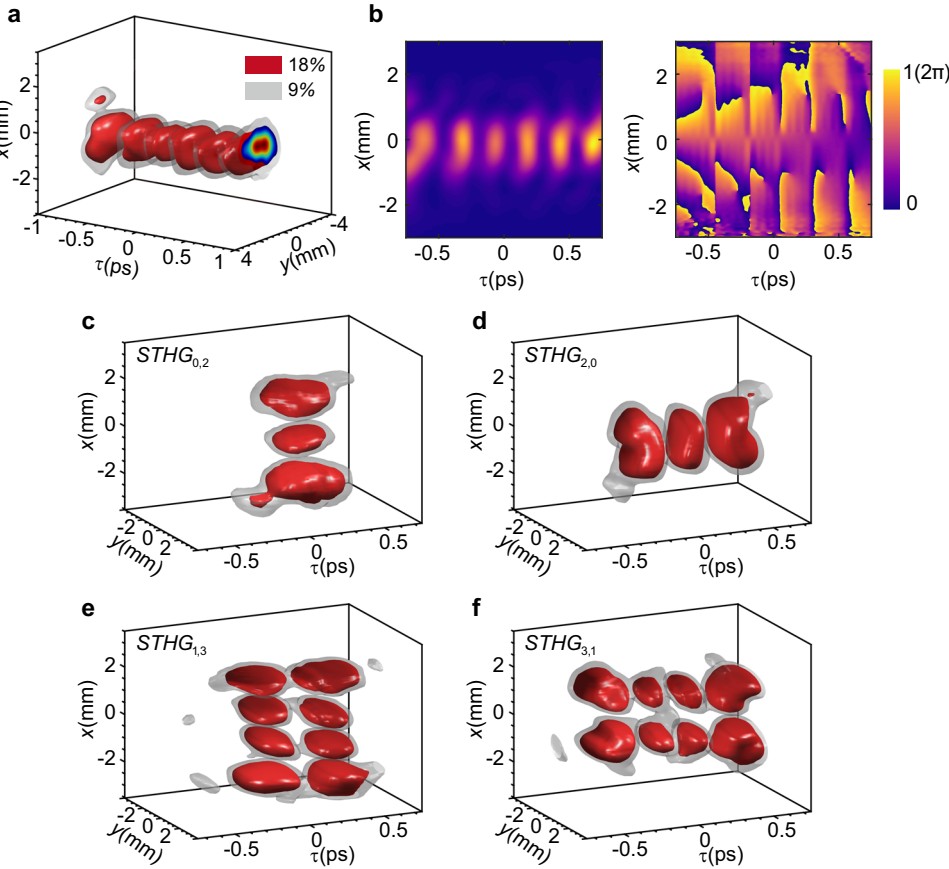

**Fig. 5 | Mode conversion of an STLG wavepacket to an STHG wavepacket by introducing spatiotemporal astigmatism. a** Experimentally reconstructed 3D iso-intensity surface of the STHG wavepacket with $p = 0$ and $l = +5$ ($STHG_{5,0}$). The maximum intensity is normalized to 1. **b** Measured accumulated spatiotemporal intensity $\int |\Psi_{STHG}(\tau,x,y)|^2 dy$ (left) and sliced phase at $y = 0$ (right) distributions.

**c**–**f** Experimentally reconstructed 3D iso-intensity surface of the STHG wavepackets with different mode indices $p$ and $l$. **c** $p = 0$ and $l = -2$; **d** $p = 0$ and $l = +2$; **e** $p = 1$ and $l = -2$; **f** $p = 1$ and $l = +2$. The spatiotemporal astigmatism strength used in the experiment is 15fs/(rad · mm).

an STHG wavepacket under different spatiotemporal astigmatism. Under a weak spatiotemporal astigmatism, the inner doughnut profile gradually deforms and the phase singularities in the center of wavepacket are separated in both space and time, while the outer ring maintains its ring structure and the phase edge dislocation is also clear. This situation also occurs for the second-harmonic STOVs in a relative thick BBO crystal[32]. As the spatiotemporal astigmatism increases, the double doughnut-shaped topology profile fully collapses leading to two lobes in space and four lobes in time and finally forming a HG profile. Supplementary Movie 2 shows a detailed numerical simulation for the topological evolution of this complex process in space-time. The experimental results agree with the simulated results very well.

It should be noted that the spatiotemporal intensity distributions of the converted STHG wavepackets are de-coupled in space and time as confirmed by the theory in the supplementary section 5, which implies that we can operate the spatial and temporal dimensions separately. In Fig. 6a, we plot the spatial intensity projections of the converted STHG wavepackets for different radial $p$ and azimuthal $l$ indices via a camera of 80 ms exposure time. Under positive spatiotemporal astigmatism, the spatial intensity projection of such STHG wavepacket exhibits a low-order spatial HG profile coupled with $M = p + |l|$ zeros (TEM$_{M0}$ mode) for negative $l$ and $N = p$ zeros (TEM$_{N0}$ mode) for positive $l$ along $x$ direction, as marked by the dark dot lines. For the negative spatiotemporal astigmatism, the above results are reversed as illustrated in Fig. 6b. Thus, we can conclude that for $\mu \cdot l < 0$, there are $M = p + |l|$ dark gaps and for $\mu \cdot l > 0$, there are $N = p$ dark gaps. This relationship undoubtedly enables us to recognize the

sign and value of radial and azimuthal quantum numbers of STLG wavepackets by introducing spatiotemporal astigmatisms, as depicted in Fig. 6c, in which the azimuthal quantum number equals $N - M$ and the radial quantum number equals the minimum of $\{M,N\}$. These results will further advance the applications of STLG wavepackets, particularly those require fast responses, such as telecommunications and data encoding where quantum numbers of STLG pulses may play important roles.

## Discussions

In conclusion, we have achieved the experimental generation of spatiotemporal Laguerre-Gaussian (STLG) wavepackets with defined radial and azimuthal indices. These elegant solutions to the scalar wave equation possess captivating features: an intrinsic transverse orbital angular momentum (OAM) and cylinder-shaped phase dislocations within their spatiotemporal profile. Our experiment successfully synthesized these wavepackets by meticulously sculpting the spatial-spectral field of a pulsed beam using a 2D pulse shaper. The resulting STLG wavepackets exhibit both intensity and phase profiles consistent with the Laguerre-Gaussian form, demonstrating high mode purity in both radial and azimuthal directions. Furthermore, we showcase the ability to transform an STLG wavepacket into a spatiotemporal Hermite-Gaussian (STHG) wavepacket through the introduction of a controlled spatiotemporal astigmatism. Both STLG and STHG sets form complete bases in the space-time domain, laying a crucial foundation for future advancements in ST-structured light engineering.

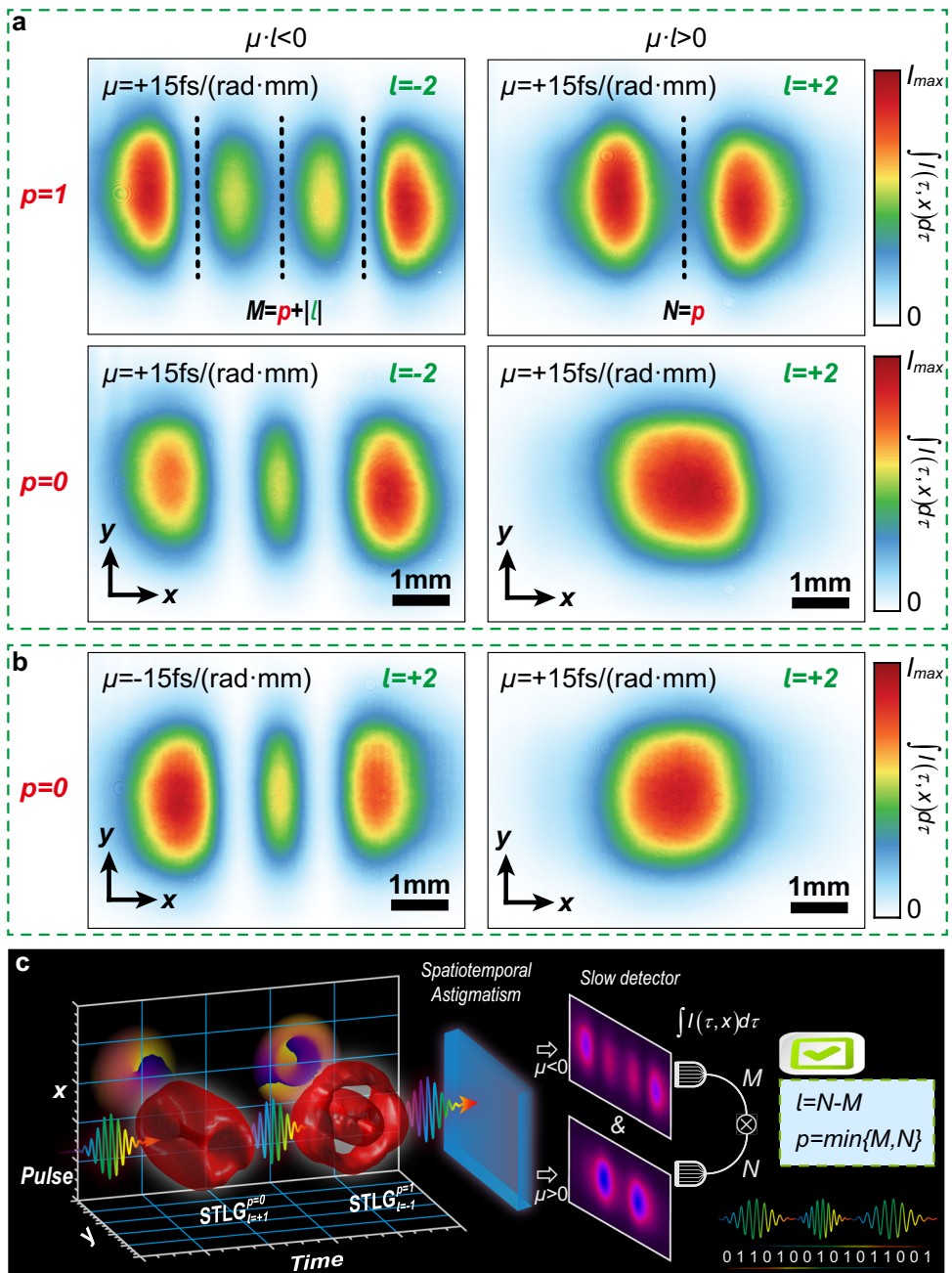

**Fig. 6 | Spatial projections for intensity profiles of STHG wavepackets via temporal integral and a scheme for recognizing quantum numbers (*p* and *l*) of STLG wavepackets. a** Comparison of spatial projection for intensity profiles of STHG wavepackets for inverse signs of azimuthal quantum number. **b** Comparison of spatial projection for intensity profiles of STHG wavepackets for inverse signs of spatiotemporal astigmatism. **c** Scheme for recognizing radial quantum number and azimuthal quantum number of STLG wavepackets.

The versatility of high-order STLG wavepackets, characterized by their non-zero radial and azimuthal indices, is truly captivating. The radial quantum number (*p*) unlocks two intriguing properties: a composite transverse OAM, composed of oppositely directed components, and a nested spatiotemporal topology within the wavepacket's profile. This unique combination makes STLG wavepackets attractive candidates for exploring complex photonic topologies[23,24,54], enabling advanced information transfer and even quantum entanglement[49–52]. However, our work also highlights a new challenge: the creation and sorting of ultra-high-order STLG wavepackets[47,55]. Overcoming this hurdle will be crucial for further unlocking the full potential of these fascinating light forms. Moreover, while our protocol utilizes both phase and amplitude modulations, which inevitably leading to some

energy loss, similar to the recent development in STOV generation[56], this drawback could be mitigated by implementing novel nanophotonic devices in the future. Finally, we believe our research paves the way for exciting explorations of STLG wavepackets beyond the realm of optics. Future investigations could extend these principles to other wave systems and nanophotonic platforms, opening up a vast landscape of possibilities.

## Methods

### Precise spatial-spectral complex-amplitude modulation based on holographic

The object pulsed beam (has a spatial Gaussian profile with a 1/*e* radius of 2 mm and a spectral bandwidth of Δλ ≈ 20nm centered on a wavelength

of 1030 nm, see Supplementary Section 7 for pulse characterization) incidents at an angle of $\alpha = 46°$ to a reflective blaze grating ($N = 1200$ lines mm$^{-1}$). The $m = +1$ order diffractive light is horizontally dispersed at a spectral resolution of $\Delta\alpha / \Delta\lambda = mN / \cos\alpha = 0.0799°$/nm and then is collimated via a cylindrical lens (focal length 10 cm) onto the SLM screen ($3840 \times 2160$ pixels with a pitch of $3.74\,\mu$m) at a spatial range of $\Delta L = 2f \tan \Delta\alpha / 2 = 2.8$mm. The horizontal spectrum resolution on the SLM screen can be calculated to be $-4.76e^{-5}$rad/fs/pixel. As a consequence, we can define the spatial-spectral coordinate $(\xi, \Omega)$ on the SLM plane. The origin of this coordinate can be further determined from the camera and spectrometer by applying a $0 - \pi$ step phase on SLM along vertical and horizontal directions, respectively. Finally, the complex amplitude of Eq. (2) can also be defined quantitatively in the experiment. The spatial-spectral complex-amplitude of an incident pulsed beam can be precisely controlled by a phase-only hologram, which has a form of

$$\psi(\Omega, \xi) = \mathrm{mod}\left\{ \mathrm{Arg}[\Psi] + g\xi \cdot \mathrm{asinc}[1 - |\Psi|] + \pi \cdot \mathrm{asinc}(|\Psi|) + \mathrm{GDD} \cdot \Omega^2, 2\pi \right\} \tag{4}$$

where $g$ denotes the frequency of a linear phase ramp, the depth of which is contingent upon the modulus of the on-demand mode. As such, the undesired spatial-spectral energy is diffracted away from the optical axis via this phase grating after propagating a short distance from the SLM (Supplementary Section 6). More detailed amplitude modulation performance of Eq. (4) can be explored in refs. 57,58. Furthermore, to suppress the residual modulation, the phase-only SLM must be calibrated to a linear $2\pi$ phase response over all 256 gray levels at wavelength 1030 nm. To produce high-quality STLG wavepackets, the pulsed beam's spatial-spectral bandwidths should entirely cover the LG pattern [Eq. (2)] loaded on the SLM. Hence, the spatial-spectral bandwidths of input pulsed beam and the spatial resolution of the SLM (or phase hologram) jointly impose a constraint on the generation of an ultrahigh-order STLG wavepacket.

## Spatiotemporal phase reconstruction

We utilized the off-axis interference scheme to reconstruct the intensity and phase information of the synthesized ST wavepackets, which is illustrated in detail in Supplementary Section 8. It is noteworthy to mention that the positions of fringes exhibit irregular shifts in jointed slice frames, attributed to the inevitable instability of interferometer caused by the motorized stage and optical misalignment, resulting in the phase random fluctuations at various time delays. Therefore, we assume the temporal phase at a specific position along $\tau$-axis is a constant, i.e., we normalize each sliced phase profile with respect to a specific spatial position $x = x_0$. For STLG, the position $x_0$ should be chosen far away from its outer ring because STLG wavepacket has circular $\pi$ phase jumps in space-time. But for STHG, it has $\pi$ phase jumps parallel to $x$-axis in space-time, this method is still inaccurate although we can roughly recognize the temporal phase dislocation in Fig. 5b. In addition, both the 2D accumulated amplitude and phase profiles have been further refined for better visualization by using a chirped $z$-transformation algorithm.

## Data availability

All other data are available in the article and its supplementary files or from the corresponding authors upon request.

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

## Acknowledgements
We acknowledge financial support from the National Key Research and Development Program of China (2022YFA1404800 [Y.C], 2019YFA0705000 [Y.C.]), National Natural Science Foundation of China (92050202 [Q.Z.], 12192254 [Y.C.], 92250304 [Y.C.]), National Research Foundation of Korea (2022R1A2C1091890 [A.C.]). Q.C. also acknowledges support by the Shanghai Sailing Program (21YF1431500). A.C. also acknowledges support by the Learning & Academic research institution for Master's·PhD students, and Post-docs Program of the National Research Foundation of Korea (RS-2023-00301938). Q.Z. also acknowledges support by the Key Project of Westlake Institute for Optoelectronics (Grant No. 2023GD007).

## Author contributions
X.L. and Q.Z. proposed the original idea and initiated this project. X.L. completed the theory and simulations. X.L. and Q.C. designed and performed the experiments. X.L. Q.C. and N.Z. both analyzed the data. X.L., Q.C., A.C., and Q.Z. both prepared the manuscript. Y.C. and Q.Z. supervised the project. All authors contributed to the discussion and writing of the manuscript.

## Competing interests
The authors declare no competing interest.
