## [Peer Review File · Nature Communications]

Spatiotemporal optical vortices with controllable radial and azimuthal quantum numbersREVIEWER COMMENTS

Reviewer #1 (Remarks to the Author):

The paper "Spatiotemporal optical vortices with controllable radial and azimuthal quantum numbers" addresses the timely research field of spatio-temporal light structuring. I think that the manuscript introduces a new aspect in the field, well supported by experimental data. Authors demonstrate the simultaneous control of both azimuthal and radial indices in OAM wavepacket structuring. I would recommend publication of this work provided that a few aspects have been further clarified:

1) The shown values of L and p indices are small, 1 or 2. In wavepacket structuring L index can be tens or more. What is the limitation related to combining the simultaneous structuring of both indices? What are the highest values that can be realized? What are the limitations on the values difference between L and P ? I would expect both an intrinsic limitation due to the lack of overlapping of the modes and a practical limitation related to the realization with an SLM. I would appreciate such aspects discussed in the paper to present the limitations in any practical application.

2) I would also appreciate the authors further clarify the spatio-temporal characteristics of the produced beam:

- Spatially, they need to actually examine the beam and find the plane where the design structure form. How can this position be tuned and predicted by design? More importantly, could other OAM beams, allow relaxing such restriction?

- Temporally, even though the pulse width is small, the temporal extension of the beams is of a few ps. This may be intentional, to have enough temporal resolution in the analysis. However, I find such aspect not discussed enough and I would appreciate a better clarification. Also, how can the temporal extension be tuned and what to expect in case of larger L and p values?

Reviewer #2 (Remarks to the Author):

In this study, Xin Liu and colleagues present a theoretical and experimental exploration of spatiotemporal Laguerre-Gaussian (STLG) wavepackets, demonstrating control over radial and azimuthal quantum numbers. Spatiotemporal optical vortices (STOVs) represent a recently identified class of structured light, characterized by their transverse orbital angular momentum (OAM) perpendicular to the direction of propagation. Predominantly, research on STOVs has been concentrated on Gaussian-type wavepackets. This work elevates the investigation of STOVs by generating STLG wavepackets with controllable radial and azimuthal quantum numbers, marking a significant advancement in STOV research. It shows compelling theoretical and experimental evidence of STLG wavepackets, offering considerable interest for both fundamental physics and photonic engineering. The paper is well-structured and logically coherent. I would like to recommend the publication of it in Nature Communications.

Technical Comments:

1. Can authors provide tables or diagrams to clearly delineate the main physical principles and the optical pathway differences between the generation of Gaussian-type STOV wavepackets and STLG wavepackets?

2. What makes the creation and sorting of ultra-high-order STLG wavepackets a challenge with the current optical setup, and what are the potential solutions?

Reviewer #3 (Remarks to the Author):

The manuscript "Spatiotemporal optical vortices with controllable radial and azimuthal quantum numbers" studies optical spatiotemporal vortices with radial and azimuthal quantum numbers, which are called spatiotemporal Laguerre-Gaussian wavepackets. Moreover, the mode conversion between an spatiotemporal Laguerre-Gaussian wavepacket and an spatiotemporal Hermite-Gaussian wavepacket is demonstrated through the application of strong spatiotemporal astigmatism, which is used for recognition of spatiotemporal Laguerre-Gaussian wavepacket with various p and l . This work can be regarded as an incremental one based on many previous works on spatialtemporal vortices. However, the novelty and avances of this work are very limited, which prevents its consideration by Nature Communications dedicated to publishing high-quality research with important advances of significance.

i) Many previous works on spatiotemporal vortices, Bessel spatiotemporal vortices, vector spatiotemporal vortices, toroidal vortices were demonstrated in the past few years. Some were listed as references of this work, while some not. Extending previous works to spatiotemporal Laguerre-Gaussian wavepackets with added radial number is not surprising.

ii) The principle and experimental setup (Fig. 3) for generating spatiotemporal optical vortices in this work are also similar to many previous works.

iii) The conversion between Laguerre-Gaussian beams and Hermite-Gaussian ones is a well-known technique. The mode conversion between an spatiotemporal Laguerre-Gaussian wavepacket and an spatiotemporal Hermite-Gaussian wavepacket based on similar principle is also not surprising.

iv) It is claimed that the demonstration offers new insights into high-dimensional quantum information, photonic topology, and nonlinear optics. Without further demonstration on these applications using spatiotemporal Laguerre-Gaussian wavepacket, the novelty is limited.

Therefore, this work on spatiotemporal optical vortices with controllable radial and azimuthal quantum numbers is suitable to a more specilized journal in optics field (Optics Express, Optics Letters) but not Nature Communications.

Response to the Reviewer Comments

Reviewer #1:

The paper "Spatiotemporal optical vortices with controllable radial and azimuthal quantum numbers" addresses the timely research field of spatio-temporal light structuring. I think that the manuscript introduces a new aspect in the field, well supported by experimental data. Authors demonstrate the simultaneous control of both azimuthal and radial indices in OAM wavepacket structuring. I would recommend publication of this work provided that a few aspects have been further clarified:

Reply: We sincerely appreciate the reviewer's positive comments and invaluable suggestions regarding our work. In the following, we clarify/address the reviewer's concerns and indicate the revisions with blue text in the revised version.

Comment 1: The shown values of L and p indices are small, 1 or 2. In wavepacket structuring L index can be tens or more. What is the limitation related to combining the simultaneous structuring of both indices? What are the highest values that can be realized? What are the limitations on the values difference between l and P? I would expect both an intrinsic limitation due to the lack of overlapping of the modes and a practical limitation related to the realization with an SLM. I would appreciate such aspects discussed in the paper to present the limitations in any practical application.

Reply: We thank the reviewer for the careful review and professional comments. We agree with the referee's insightful comment on the practical upper bound of radial and azimuthal indices. In our work, there are two aspects for the limitation of simultaneous structuring both radial and azimuthal indices. **(i)** The dependence of the spatial/spectral radius of a LG mode (described by equation (2) in the main text) on its radial and azimuthal indices. For a given beam waist w_1 , the outer radius of LG mode loaded on the SLM increases with p and l, i.e., $\sigma_{p,l} = w_1 \sqrt{2p + |l| + 1}$, the spatial and spectral bandwidths of the input pulsed beam on the SLM should fully cover this pattern. Thus, the minimum spatial/spectral bandwidths of the input pulsed beam on the SLM determines the maximum outer radius $\sigma_{p,l}$ of the LG mode. Based on our experimental conditions [a spectral width of $\sim 20\text{nm}$ (the spread width along x axis is $\sim 2.8\text{mm}$), spatial width (along y axis) of $\sim 2\text{mm}$ of the input pulsed beam and adopted beam waist of LG mode is 0.4mm], an STLG with a maximum $2p + |l| \approx 5$ could be achieved. The value difference between l and p in our present configuration should stay within the limits of the abovementioned conditions. **(ii)** For the input pulsed beam of fixed bandwidths, simultaneously increasing the indices p and l requires the reduction of the beam waist w_1 . However, smaller w_1 will demand a higher spatial resolution of SLM. Additionally, since the STLG wavepacket, presented in our work, is characterized in the far-field (free propagation around $L=1.2\text{m}$ after the pulse shaper in the experimental setup), LG mode with smaller w_1 will undergo rapid spatial spreading after the pulse shaper, requiring a larger aperture camera to record, although a spatial focus phase on the SLM or a lens can be

employed to image the STLG to mitigate the spatial spreading induced by a long distance free-propagation after the grating.

We have added the following sentences in the revised version:

“To produce high-quality STLG wavepackets, the pulsed beam’s spatial-spectral bandwidths should entirely cover the LG pattern [Eq. (2)] loaded on the SLM. Hence, the spatial-spectral bandwidths of input pulsed beam and the spatial resolution of the SLM (or phase hologram) jointly impose a constraint on the generation of an ultrahigh-order STLG wavepacket.”

“It is worth mentioning that such structured ST wavepackets are synthesized in the far-field via a time-delayed spatial propagation after the grating, which inevitably leads to a spatial/temporal broadening. The spatial and temporal width of the STLG wavepackets are determined by the time-dispersion and space-diffraction phases (see equation (21) in the Supplementary Section 2). Hence, we can regulate their spatial and temporal widths by manipulating the diffraction and dispersion phases on the SLM.”

Comment 2: I would also appreciate the authors further clarify the spatio-temporal characteristics of the produced beam:

- Spatially, they need to actually examine the beam and find the plane where the design structure form. How can this position be tuned and predicted by design? More importantly, could other OAM beams, allow relaxing such restriction?
- Temporally, even though the pulse width is small, the temporal extension of the beams is of a few ps. This may be intentional, to have enough temporal resolution in the analysis. However, I find such aspect not discussed enough and I would appreciate a better clarification. Also, how can the temporal extension be tuned and what to expect in case of larger L and p values?

Reply: We thank the reviewer for the careful reading and valuable comments.

(i) Spatially - In our proposed scheme, the desired ST wavepacket is generated through the following 2D ST Fresnel diffraction, as displayed in Supplementary Section 2:

$$\Psi(t, x, L) \propto \iiint \Psi(\Omega, \xi, 0) e^{ik_0 \frac{\xi^2}{2L} - i\alpha\Omega^2} e^{-ik_0 \frac{x\xi}{L} - i\Omega t} d\Omega d\xi.$$

The position of the plane L determines two common cases: near-field and far-field. The near-field case is a mapping relation between the input and output field, and the far-field case is a Fourier transformation relation between the input and output field. For STLG wavepackets, the synthesized plane L is not restricted since it is a stable solution of the scalar paraxial wave equation (as shown in the Supplementary Section 2), we only need to control the group delay dispersion (GDD) $\alpha = -\gamma^2 k_0 / 2L$ of time-frequency to ensure a balance between temporal dispersion and spatial diffraction (the first phase term in the integral). In our experimental setup, we chose the plane at around 1.2m behind the pulse shaper for two reasons: (1) To ensure that the optical path length between the reference and the object is nearly identical. (2) To ensure that the undesired diffraction orders from the complex-amplitude hologram are directed away from the optical axis. For other OAM beams or ST-structured wavepackets, the choice of plane L depends on the input mode, for example, the case in ref.[*Nat Commun* 13, 4021 (2022)] belongs to the near-field situation while the case in ref.[*Nat. Photonics* 14, 350–354 (2020)] belongs to the far-field situation.

(ii) Temporally – In our experiment setup, the STLG wavepacket is synthesized at the plane $L=1.2\text{m}$ after pulse shaper, we add a GDD phase on the SLM for balancing the spatial diffractive phase introduced by propagation. Consequently, the generated wavepacket is broadened in the temporal domain. As the reviewer mentioned, this also facilitates the 3D wavepacket reconstruction during analysis. On the other hand, as shown in equation (21) in the Supplementary Section 2, we can also tune their ST-width by simultaneously controlling the second-order phases (focus or defocus) on the SLM in the 2D spatial-spectral coordinates. For larger p and l values, the ST-width will be broadened, and we can add a focused phase on the SLM to reduce their broadening.

To clarify, we have added the following sentences in the revised version.

“The ST wavepacket is synthesized in the far field via a time-delayed spatial propagation after the grating. Actually, we can readily realize a near field ST-wavepacket engineering by reducing the diffraction and dispersion phases.”

“The spatial and temporal width of the STLG wavepackets are determined by the time-dispersion and space-diffraction phases (see equation (21) in the Supplementary Section 2). Hence, we can regulate their spatial and temporal widths by manipulating the diffraction and dispersion phases on the SLM.”

Reviewer #2:

In this study, Xin Liu and colleagues present a theoretical and experimental exploration of spatiotemporal Laguerre-Gaussian (STLG) wavepackets, demonstrating control over radial and azimuthal quantum numbers. Spatiotemporal optical vortices (STOVs) represent a recently identified class of structured light, characterized by their transverse orbital angular momentum (OAM) perpendicular to the direction of propagation. Predominantly, research on STOVs has been concentrated on Gaussian-type wavepackets. This work elevates the investigation of STOVs by generating STLG wavepackets with controllable radial and azimuthal quantum numbers, marking a significant advancement in STOV research. It shows compelling theoretical and experimental evidence of STLG wavepackets, offering considerable interest for both fundamental physics and photonic engineering. The paper is well-structured and logically coherent. I would like to recommend the publication of it in Nature Communications.

Reply: We sincerely appreciate the reviewer’s encouraging comments and invaluable suggestions. In the following, we clarify/address the reviewer’s concerns and indicate the revisions with blue text in the revised version.

Technical Comments:

Comment 1: Can authors provide tables or diagrams to clearly delineate the main physical principles and the optical pathway differences between the generation of Gaussian-type STOV wavepackets and STLG wavepackets?

Reply:

Fig. R2 | a. Principle for the generation of Gaussian-type STOV wavepackets. **b.** Principle for the generation of STLG wavepackets.

We thank the reviewer for the careful reading and this kind suggestion. In the original manuscript, we may have not adequately illustrated the features of our experimental setup in Fig. 3 for simplicity purposes. The primary distinction between the generation of Gaussian-type STOV wavepackets and STLG wavepackets lies in their modulation techniques inside pulse shaper. In the setups for STOVs generation, STOVs are produced via phase-only

modulations, relying on the conservation of phase singularities under a 2D Fourier transformation. Consequently, this approach allows the control of only one single degree of freedom for Gaussian-type STOVs, as shown in Fig. R2a. In our work, we adopted a complex-amplitude modulation, which allows for simultaneous modulation of both amplitude and phase [see Fig. R2b], granting precise control over multiple degrees of freedom for ST wavepacket synthesis. Briefly, in this scheme, the input field is tailored by a complex-amplitude modulation hologram in spatial-spectral domain and the desired ST-wavepacket is produced in specific diffraction order. By filtering out the undesired diffraction orders, the STLG wavepacket is generated at the output plane. In our experimental setup [see Fig. R2b or Fig. 3 and Supplementary Section 6 in the revised version], we chose the on-axial zeroth diffraction order to avoid spatial dispersion induced by optical path deflection inside the holographic pulse shaper, as described in methods section of the main text.

To make this distinction more clear, we have revised Fig.3a and added corresponding sentences in the revised revision as:

“In this scheme, the desired ST wavepacket, as denoted by Eq. (2), is produced by selecting the zeroth order diffraction to circumvent spatial dispersion caused by diffractive deflection angles (see Methods and Supplementary Section 6).”

“The undesired diffractive orders after the grating are blocked by an iris and only the zeroth order is selected and detected by the camera.”

Comment 2: What makes the creation and sorting of ultra-high-order STLG wavepackets a challenge with the current optical setup, and what are the potential solutions?

Reply: In our work, the bandwidth of pulsed beam and spatial resolution of SLM are the major limitations for creating a high-order STLG wavepacket in our current optical apparatus. The reasons are as the following: (1) The LG mode loaded on the SLM has an outer radius of $\sigma_{p,l} = w_1 \sqrt{2p + |l| + 1}$. To produce a high-quality STLG wavepacket, the spectrum of input pulse must entirely cover this LG mode profile. (2) When considering a given spectral bandwidth, the increase mode numbers of STLG wavepacket necessitates a reduction in beam waist w_1 of LG mode loaded on the SLM. This reduction in w_1 results in less sampling of the SLM on the hologram. Consequently, it would be beneficial to achieve ultra-high order STLG wavepackets by replacing the SLM with nanophotonic devices such as metasurface [*Nat Commun* 14, 6410 (2023)]. Implementing efficient and precise mode sorting of STLG wavepackets is challenging due to their spatiotemporal coupling and ultrafast temporal profile. Combining the proposed spatiotemporal mode conversion between STLG wavepacket and STHG wavepacket with some mature approaches [*Nat. Commun.* 10, 1865 (2019), *Phys. Rev. Lett.* 119, 263602 (2017) and *Phys. Rev. Lett.* 130, 240801 (2023)] may provide promising solutions for their sorting.

Reviewer #3:

The manuscript "Spatiotemporal optical vortices with controllable radial and azimuthal quantum numbers" studies optical spatiotemporal vortices with radial and azimuthal quantum numbers, which are called spatiotemporal Laguerre-Gaussian wavepackets. Moreover, the mode conversion between an spatiotemporal Laguerre-Gaussian wavepacket and an spatiotemporal Hermite-Gaussian wavepacket is demonstrated through the application of strong spatiotemporal astigmatism, which is used for recognition of spatiotemporal Laguerre-Gaussian wavepacket with various p and l . This work can be regarded as an incremental one based on many previous works on spatialtemporal vortices. However, the novelty and avances of this work are very limited, which prevents its consideration by Nature Communications dedicated to publishing high-quality research with important advances of significance.

Reply: We thank the reviewer for taking time to review this manuscript. However, we believe the reviewer may have missed the key points of the current work and we could not agree with the reviewer's assessment on the novelty and pimportance of our work as reviewers #1 and #2 have recognized. In the following, we clarify/address the reviewer's concerns and indicate the revisions with blue text in the revised version.

Comment 1: Many previous works on spatiotemporal vortices, Bessel spatiotemporal vortices, vector spatiotemporal vortices, torioidal vortices were demonstrated in the past few years. Some were listed as references of this work, while some not. Extending previous works to spatiotemporal Laguerre-Gaussian wavepackets with added radial number is not surprising.

Reply: We could not agree with the reviewer's assessment that generation of spatiotemporal Laguerre-Gaussian wavepackets with added radial number is not surprising. The reviewer is certainly correct that many works on spatiotemporal vortices have been reported recently. However, this itself demonstrates the importance and strong interests in generation spatiotemporal wavepackets with various different characteristics. We want to emphasize here that the current work is not merely an extension of the previous works on spatiotemporal vortices. In the spatiotemporal vortices studies the reviewer mentioned, the spatiotemporal structures were generated through phase-only modulation. However, achieving STOVs with dual controllable degrees of freedom incorporating both radial and azimuthal indices remains to be challenging for these previous phase-only modulation techniques as it requires simultaneously performing prescribed amplitude and phase modulations in the spatiotemporal domain, which is not trivial. In this work, we successfully synthesized such advanced and novel STLG with dual controllable mode numbers by employing a spatiotemporal complex-amplitude modulation approach. The complex-amplitude modulation scheme with a holographic pulse shaper allows for simultaneous control over radial and azimuthal indices of such STLG wavepackets, something that is not possible with previous methods of generating STOVs.

Comment 2: The principle and experimental setup (Fig. 3) for generating spatiotemporal optical vortices in this work are also similar to many previous works.

Reply:

Fig. R3 | Our modulation scheme for generating STLG wavepackets.

We strongly disagree with the reviewer on this statement. As we stated above, the experimental setup used in this work may appear similar to those 4-f pulse shapers used in previous works, the modulation scheme and working principle are actually very different. For simplicity purpose we may have not adequately illustrate the features of our experimental setup in Fig. 3 in the original version. The principle for generating STLG in our work is based on complex-amplitude modulation strategy [see Fig. R3]. Unlike the phase-only modulation methods in previous works, this innovative method allows for simultaneous modulation of both amplitude and phase [see Fig. R3] in the spatiotemporal domain, granting precise control over multiple degrees of freedom for ST wavepackets synthesis. Briefly, in this scheme, the desired ST-wavepacket is tailored by a digital hologram and is encoded in specific diffraction order and must be filtered by an iris [see Fig. 3 and Supplementary Section 6 in the revised version]. In our experimental setup, we choose the zeroth diffraction order to avoid spatial dispersion induced by optical path deflection inside the holographic pulse shaper through the designed hologram given as follows

$$\psi(\Omega, \xi) = \text{mod}\{\text{Arg}[\Psi(\Omega, \xi)] + g\xi \cdot [1 - |\Psi(\Omega, \xi)|] + GDD \cdot \Omega^2, 2\pi\},$$

where g denotes the frequency of a linear phase ramp, the depth of which is contingent upon an inverse Sine cardinal function of the modulus of the on-demand mode. We have detailly analyzed the complex-amplitude modulation performance of this equation in ref.[53] of the main text. Hence, the experimental principle in our work is very different from previous methods.

For further clarify this point, we have revised Fig.3, its caption and corresponding sentences in the revised version.

Comment 3: The conversion between Laguerre-Gaussian beams and Hermite-Gaussian ones is a well-known technique. The mode conversion between an spatiotemporal Laguerre-Gaussian wavepacket and an spatiotemporal Hermite-Gaussian wavepacket based on similar principle is also not surprising.

Reply: Again, we strongly disagree with the reviewer's assessment that "mode conversion between an spatiotemporal Laguerre-Gaussian wavepacket and an spatiotemporal Hermite-Gaussian wavepacket based on similar principle is also not surprising". The reviewer is certainly correct that the conversion between Laguerre-Gaussian beams and Hermite-Gaussian ones is a well-known technique in the spatial domain. However, mode conversion between STLG wavepackets and STHG wavepackets is achieved in the spatiotemporal domain in our manuscript. To the best of our knowledge, this achievement marks the first

instance of successfully attaining such spatiotemporal modes conversion in the spatiotemporal domain. Prior to our demonstration of these novel wavepackets and revealing their mode conversion properties, no one would have even thought about the mode conversion between them. One should not conclude that the demonstration of this mode conversion in the spatiotemporal domain is trivial or not surprising, just because something is well known in the spatial domain.

This point can be further illustrated by the recent rapid advances made in spatiotemporal wavepackets. For examples, the spatial optical vortices have been well-studied before the experimental realizations of STOVs [*Optica* 6, 1547 (2019) and *Nat. Photonics* 14, 350–354 (2020)]. After that, the spatiotemporal vortices have also gradually extended into other waves, such as acoustics and electrons [*Nat. Commun* 14, 6238 (2023), *Phys. Rev. A* 107, L031501 (2023), *Phys. Rev. Lett.* 131, 014001 (2023) and *Phys. Rev. Lett.* 132, 054003 (2024), etc.]. In nonlinear frequency conversion, the second-order and high-order harmonic generations of conventional spatial optical vortices are also well-known techniques and have been extensively studied in the spatial domain, while the nonlinear effects of STOVs based on similar principles were still explored in recent years and attracted researcher's attention [*Optica* 8, 594-597 (2021), *Nat. Photon.* 15, 608-613 (2021) and *Phys. Rev. Lett.* 127, 273901 (2021)].

Furthermore, the synthesized STLG and STHG wavepackets define complete and orthonormal bases for ST structured lights. The converted STHG wavepacket exhibits an ST-decoupled intensity distribution, which can be used to reveal mode indices information of ST-coupled STLG ultrafast wavepackets. Thus, such a mode conversion in the spatiotemporal domain can be used to solve one of the challenges in potential application of spatiotemporal vortices. And by the way, it has come to our attention that, after our preprint was posted on arXiv and during the subsequent peer review process, a similar work about ST structured lights (including STLG and STHG, etc.) was also reported in arXiv from another group [*arXiv preprint arXiv:2402.07794* (2024)].

Hence, we strongly believe that one shouldn't deny the novelty and potential values of such mode conversion in the spatiotemporal domain just because similar conversion is well known in the conventional spatial domain. The physics is very different despite the mathematical similarity.

Comment 4: It is claimed that the demonstration offers new insights into high-dimensional quantum information, photonic topology, and nonlinear optics. Without further demonstration on these applications using spatiotemporal Laguerre-Gaussian wavepacket, the novelty is limited.

Reply: First of all, we hope that by this point we have convinced the reviewer that the working principle and experimental methods are not simple extensions of those reported in the existing literatures on spatiotemporal vortices. The experimental demonstration of these new spatiotemporal wavepackets and their conversion are novel and non-trivial with important potential applications. And we hope the reviewer find these findings already substantial enough to justify the novelty of this manuscript.

With regards to our statement of “the demonstration offers new insights into high-dimensional quantum information, photonic topology, and nonlinear optics,” this is a natural conclusion of what we can learn from the developments of the spatial counterparts of these spatiotemporal wavepackets. The spatiotemporal Laguerre-Gaussian (STLG) optical vortices possess an intrinsic transverse orbital angular momentum (OAM) and edge phase dislocations, respectively defined by its azimuthal and radial quantum numbers in the space time domain. In the quantum realm, LG mode with dual degrees of freedom has also been applied to demonstrate the Einstein-Podolsky-Rosen correlations between down-converted photons [*Phys. Rev. Lett.* 123, 060403 (2019)] and the violation of a Bell inequality in two-dimensional state spaces [*Phys. Rev. A* 98, 042134 (2018)]. Furthermore, the LG and HG modes, serving as the basis states of structured lights, have a great number of controllable degrees of freedom, rendering them the bright candidates for quantum entanglement [*Phys. Rev. Lett.* 108, 173604 (2012), *Appl. Phys. Lett.* 124, 110501 (2024) and *AVS Quantum Sci.* 1, 011701 (2019)]. It is well known that space and time are dual analogies. Reasonably, the STLG wavepacket has promising to serve as a versatile and advanced tool for encoding quantum information due to their high-dimensional spatiotemporal degrees of freedom and can enable encoding information in higher-dimensional quantum states, leading to increased information capacity and security. Photonic counterparts of topological solitons have aroused intense interest of researchers in recent years [*Nat. Photonics* 18, 15 (2024)]. Recently, optical scalar hopfions have also been experimentally demonstrated through the spatial mapping of a STOV [*Nat. Photonics* 16, 519 (2022), *eLight* 2, 1 (2022) and *ACS Photonics* 10, 3384 (2023)]. The STLG wavepacket introduces a radial degree of freedom across both space and time, resulting in a nested spatiotemporal topology texture. This distinctive architecture facilitates the generation of intricate topologies localized in the x-y-t 3D space, offering a greater degree of control over their properties. Finally, ultrafast STOV pulses can undergo nonlinear frequency conversion processes such as harmonic generation and parametric up-conversion to modulate their photon transverse OAM [*Optica* 8, 594 (2021), *Nat. Photonics* 15, 608 (2021), *Phys. Rev. Lett.* 127, 273901 (2021) and *Phys. Rev. Lett.* 130, 153803 (2023)]. As such, the STLG with complex ST structure and dual degrees of freedom in space-time plane is naturally expected to reveal some nonlinear processes such as control over the nonlinear conversion processes and complex frequency spectra generation, etc.

Thus we hope the reviewer can understand that this is a general statement as a natural conclusion of what one can learn from the developments of the spatial counterparts of these spatiotemporal wavepackets. Continued studies on these topics by enthusiastic scientists can further promote the potential efficacy and versatility of STLG wavepackets in these fields, paving the way for practical applications and technological innovations. However, demonstration of these applications are far beyond the scope of a single manuscript.

Comment 5: Therefore, this work on spatiotemporal optical vortices with controllable radial and azimuthal quantum numbers is suitable to a more specialized journal in optics field (Optics Express, Optics Letters) but not Nature Communications.

Reply: As shown above, we have tried our best to address the reviewer’s concerns point-by-point in details. The working principle and experimental methods are not simple

extensions of those reported in the existing literatures on spatiotemporal vortices. The experimental demonstration of these new spatiotemporal wavepackets and their conversion are novel and non-trivial with important potential applications. Corresponding modifications have been made in the revised version to better clarify the novelty of our research. We hope that the above explanations satisfyingly address the reviewer's concerns and the novelty of this work is now clear to the reviewer based on these facts.

REVIEWER COMMENTS

Reviewer #1 (Remarks to the Author):

The authors have addressed all my main comments.

Reviewer #2 (Remarks to the Author):

The authors have fully addressed my comments, rendering the paper suitable for publication from my perspective.

Reviewer #3 (Remarks to the Author):

I appreciate the efforts that the authors have made in response to the questions and concerns. However, after carefully reading the authors' response file and revised paper, I do not think the novelty and advances of the revised paper are improved to meet the high publication standard of Nature Communications.

1. Novelty of spatiotemporal Laguerre-Gaussian (LG) wavepacket:

In the area of spatiotemporal optical vortices, the current work is indeed just an extension of previous works. There is no fundamental innovation in the generation of this so called spatiotemporal Laguerre-Gaussian (LG) wavepacket. Even if the authors claim again that the demonstrated work implements spatiotemporal optical vortices with dual controllable degrees of freedom incorporating both radial and azimuthal indices, the basic principle to synthesize the spatiotemporal optical vortices based on pulse shapers (diffraction grating, cylindrical lens, 2D SLM) is exactly the same as previous works (<https://doi.org/10.1038/s41566-020-0587-z>). The authors published the first paper of spatiotemporal optical vortices on Nature Photonics in 2020 (<https://www.nature.com/articles/s41566-020-0587-z>). After that, they also demonstrated spatiotemporal vortices with tilted OAM, published on National Science Review (<https://doi.org/10.1093/nsr/nwab149>), spatiotemporal optical vortex with singularities embedded in multiple domains, published on Chinese Optics Letters (<https://opg.optica.org/col/abstract.cfm?uri=col-21-8-080003>), Bessel spatiotemporal optical vortices, published on Science Bulletin (<https://www.sciencedirect.com/science/article/pii/S2095927321005120>), and cylindrical vector spatiotemporal optical vortices, published on Nanophotonics (<https://doi.org/10.1515/nanoph-2021-0427>). These previous works can be regarded as the extension of their early Nature Photonics paper. In general, different spatially sculpted light beams could be also extended to their spatiotemporal counterparts following the similar principle, and Laguerre-Gaussian one is just another example. There are lots of spatially sculpted light beams such as optical vortex, Ince-Gaussian beam, Airy beam, pin beam, Hermite-Gaussian beam, Laguerre-Gaussian beam, Bessel beam, vector beam, vector vortex, and some of them also have multiple degrees of freedom like HG beam, LG beam, vector beam and vector vortex beam. Should we publish multiple Nature Communications papers by implementing their spatiotemporal counterparts? I believe the answer is definitely negative. The authors have already demonstrated Bessel and vector spatiotemporal vortices before, where the implementation challenge is even greater than the current work of spatiotemporal LG wavepacket. Therefore, without adding new mechanism for the spatiotemporal wavepacket synthesizer or showing its advances in practical applications, I insist that the novelty of the current work is low, which does not meet the publish standard of a high-impact journal like Nature Communications.

2. Novelty of complex-amplitude modulation approach:

In the response file, the authors clarify that they employ a complex-amplitude modulation approach. However, after going through the details in Fig. 3 of the revision, you can find the authors still employ the phase-only SLM. It is obvious and has been widely studied to implement complex-amplitude modulation with computer-generated hologram (CGH) embedded on the SLM (see [1] Appl. Opt. 38, 5004-5013 (1999); [2] <https://opg.optica.org/josab/abstract.cfm?uri=josab-24-12-2940>; [3] <https://www.nature.com/articles/srep15426>; [4] <https://opg.optica.org/ol/fulltext.cfm?uri=ol-34-1-34&id=175535>). Even the LG beam generation by the complex-amplitude modulation has been extensively studied. The second reviewer also raised the concern about the difference in the experiment between previous works of spatiotemporal optical vortices and this work of spatiotemporal LG wavepacket. In the response to the comment 1 of the second reviewer, it is clear from Fig. R2a and R2b the experimental implementation is quite similar except for the CGH-based complex-amplitude modulation and iris employed in the current work, which also increase the complexity. Furthermore, the digital CGH method inducing multiple diffraction orders also causes large amount of power loss after choosing a specific diffraction order by the iris. Giving the fact that the complex-amplitude modulation approach is obvious with increased complexity and loss, simply incorporating it into the spatiotemporal LG wavepacket generation shows limited novelty.

3. Novelty of mode conversion between spatiotemporal LG wavepacket and an spatiotemporal HG wavepacket:

In the response file, the authors also agree that the conversion between LG beams and HG ones is a well-known technique (see [1] Phys. Rev. A 45, 335 8185-8189 (1992) [2] Nat. Commun. 10, 1865 (2019)). It is indeed not surprised the similar conversion works for spatiotemporal LG wavepacket and spatiotemporal HG wavepacket. No new physics is found in this mode conversion.

4. Similar works of spatiotemporal wavepackets beyond spatiotemporal optical vortices:

The current work combines LG beam with spatiotemporal wavepacket, which is an extension of previous spatiotemporal optical vortices. This is similar to many previous works in this area such as spatiotemporal Bessel wavepacket, spatiotemporal vector wavepacket, spatiotemporal sculpted (tilted OAM, multiple singularities) wavepacket. All these can be regarded as extension works of the early spatiotemporal optical vortices. The current work of spatiotemporal LG wavepacket shows multi-ring topology in its spatiotemporal profile, which can be also seen in the spatiotemporal Bessel wavepacket demonstrated before (<https://www.sciencedirect.com/science/article/pii/S2095927321005120>). In particular, even for the spatiotemporal LG wavepacket, its similar idea has been proposed and studied in another previous work published on Optics Express (<https://doi.org/10.1364/OE.447487>).

5. When carefully reading the responses to all reviewers, we found the reviewer 1's comment on the limitation of l and p values for STLG wavepackets and reviewer 2's comment on the physical difference between the generation of Gaussian-type STOV wavepackets and STLG wavepackets could help to improve the quality of the work. However, without adding new results to improve the work, especially for the limitation of l and p values, the technical advances of the revision are limited.

6. The authors disagree with the previous comment on the new insights into high-dimensional quantum information, photonic topology, and nonlinear optics. However, without adding supported results, how can the authors conclude the advantages of the current work of spatiotemporal LG wavepackets with controllable radial and azimuthal quantum numbers? If the basic principle and technique for synthesizing spatiotemporal LG wavepacket are similar to previous works of spatiotemporal optical vortices, one would expect advanced specific applications with the generated spatiotemporal LG wavepackets. Otherwise, the novelty and advances of the current work are not high enough to be published on Nature Communications.

In a word, the authors proposed and demonstrated spatiotemporal optical vortices with controllable radial and azimuthal indices. This work is indeed an incremental one with limited novelty compared to previous works in terms of fundamental principle of spatiotemporal wavepacket generation, complex-amplitude modulation and LG-HG mode conversion. The advances of the current work

also need improvement in terms of limited values of radial and azimuthal indices (l , p) and specific practical application. Moreover, the authors do not fully and seriously respond to the comments of the reviewers and it is pity the novelty and advances of the revision do not significantly improve compared to the original version. Just providing some explanations without deep level improvement is not convincing. Therefore, this paper may be published on a more specified journal showing incremental progress of this area, but not Nature Communications aiming at publishing original and high-impact works.

Response to the Reviewer Comments

Reviewer #1 (Remarks to the Author):

The authors have addressed all my main comments.

Reply:

We deeply appreciate the recognition of the reviewer regarding the revision and explanation of the work.

Reviewer #2 (Remarks to the Author):

The authors have fully addressed my comments, rendering the paper suitable for publication from my perspective.

Reply:

We deeply appreciate the recognition of the reviewer regarding the revision and explanation of the work.

Responses to Comments by Reviewer #3

I appreciate the efforts that the authors have made in response to the questions and concerns. However, after carefully reading the authors' response file and revised paper, I do not think the novelty and advances of the revised paper are improved to meet the high publication standard of Nature Communications.

Reply:

We thank the reviewer for recognizing our efforts in responding the questions and concerns. However, we could not agree with the reviewer's assessment on the novelty and advances of our work. We believe that most of the issues raised by reviewer #3 are due to misunderstanding or sometimes confusions between the spatially and spatiotemporally structuring of light. In the following, we give our point-by-point responses to the reviewer's comments in blue text.

Comment 1:

1. Novelty of spatiotemporal Laguerre-Gaussian (LG) wavepacket:

In the area of spatiotemporal optical vortices, the current work is indeed just an extension of previous works. There is no fundamental innovation in the generation of this so called spatiotemporal Laguerre-Gaussian (LG) wavepacket. Even if the authors claim again that the demonstrated work implements spatiotemporal optical vortices with dual controllable degrees of freedom incorporating both radial and azimuthal indices, the basic principle to synthesize the

spatiotemporal optical vortices based on pulse shapers (diffraction grating, cylindrical lens, 2D SLM) is exactly the same as previous works (<https://doi.org/10.1038/s41566-020-0587-z>). The authors published the first paper of spatiotemporal optical vortices on Nature Photonics in 2020 (<https://www.nature.com/articles/s41566-020-0587-z>). After that, they also demonstrated spatiotemporal vortices with tilted OAM, published on National Science Review (<https://doi.org/10.1093/nsr/nwab149>), spatiotemporal optical vortex with singularities embedded in multiple domains, published on Chinese Optics Letters (<https://opg.optica.org/col/abstract.cfm?uri=col-21-8-080003>), Bessel spatiotemporal optical vortices, published on Science Bulletin (<https://www.sciencedirect.com/science/article/pii/S2095927321005120>), and cylindrical vector spatiotemporal optical vortices, published on Nanophotonics (<https://doi.org/10.1515/nanoph-2021-0427>). These previous works can be regarded as the extension of their early Nature Photonics paper. In general, different spatially sculpted light beams could be also extended to their spatiotemporal counterparts following the similar principle, and Laguerre-Gaussian one is just another example. There are lots of spatially sculpted light beams such as optical vortex, Ince-Gaussian beam, Airy beam, pin beam, Hermite-Gaussian beam, Laguerre-Gaussian beam, Bessel beam, vector beam, vector vortex, and some of them also have multiple degrees of freedom like HG beam, LG beam, vector beam and vector vortex beam. Should we publish multiple Nature Communications papers by implementing their spatiotemporal counterparts? I believe the answer is definitely negative. The authors have already demonstrated Bessel and vector spatiotemporal vortices before, where the implementation challenge is even greater than the current work of spatiotemporal LG wavepacket. Therefore, without adding new mechanism for the spatiotemporal wavepacket synthesizer or showing its advances in practical applications, I insist that the novelty of the current work is low, which does not meet the publish standard of a high-impact journal like Nature Communications.

Reply:

We appreciate the reviewer for reading and citing many our previous publications. However, we could not agree with the reviewer's assessment of the novelty of our work.

First of all, we could not agree with the reviewer's **categorical rejection** of even the potential of publication of future works on spatiotemporally structured wavepackets just because their counterparts have been studied in the spatial domain. According to this reviewer, one could also say that the generation of STOV is trivial as well, because spatial optical vortex has been studied in the past 30 years and what is entailed in the generation of STOV is simply loading a vortex phase to the pulse shaper. Even more, if one follows the same rationale of this reviewer, many of the following work in SHG STOV, High-order Bessel STOV, toroidal optical vortices and pulses, acoustics spatiotemporal vortices, and so on so forth, should not have been published in Nature Photonics or Nature Communications at all (many of the follow-up works are actually from other groups around the world). We feel that such a categorical rejection even for future potential works is rather shocking. How could one determine the novelty or importance of these

potential future works even without knowing how these could be done and what these could be used for?

Moreover, we respectfully disagree with the opinion that our current work is indeed just an extension of previous research. Firstly, the STLG wavepackets reported in our study cannot be synthesized using earlier methods. Secondly, the STLG wavepackets exhibit a more intricate space-time coupled structure and multiple controllable degrees of freedom compared to conventional STOVs. Also, the STLG and STHG wavepackets generated in our research form complete and orthonormal bases, offering a wealth of mode combinations for spatiotemporal structured light. All these aspects have been clarified in our manuscript, as reviewers #1 and #2 have recognized.

In regarding to the novelty of this work, as the reviewer #3 already admitted in his or her own comment #2 (see below), we did add new mechanism for the spatiotemporal wavepacket synthesizer in order to produce these STLG wavepackets. Then we created STLG wavepackets which have never been experimentally done nor been theoretically proposed before (see below for our response to comment #4, where this reviewer refuses to recognize this point by citing irrelevant works). We also studied the mode conversion mechanism in the spatiotemporal domain that has never been thought before, a mechanism that is very different from the mode conversion in spatial modes. All of these elements are novel and should not be regarded as simple extensions of the previous work.

Comment 2:

2. Novelty of complex-amplitude modulation approach:

In the response file, the authors clarify that they employ a complex-amplitude modulation approach. However, after going through the details in Fig. 3 of the revision, you can find the authors still employ the phase-only SLM. It is obvious and has been widely studied to implement complex-amplitude modulation with computer-generated hologram (CGH) embedded on the SLM (see [1] Appl. Opt. 38, 5004-5013 (1999); [2] <https://opg.optica.org/josab/abstract.cfm?uri=josab-24-12-2940>; [3] <https://www.nature.com/articles/srep15426>; [4] <https://opg.optica.org/ol/fulltext.cfm?uri=ol-34-1-34&id=175535>). Even the LG beam generation by the complex-amplitude modulation has been extensively studied. The second reviewer also raised the concern about the difference in the experiment between previous works of spatiotemporal optical vortices and this work of spatiotemporal LG wavepacket. In the response to the comment 1 of the second reviewer, it is clear from Fig. R2a and R2b the experimental implementation is quite similar except for the CGH-based complex-amplitude modulation and iris employed in the current work, which also increase the complexity. Furthermore, the digital CGH method inducing multiple diffraction orders also causes large amount of power loss after choosing a specific diffraction order by the iris. Giving the fact that the complex-amplitude modulation approach is obvious with increased complexity and loss, simply incorporating it into the spatiotemporal LG wavepacket generation shows limited novelty.

Reply:

As we mentioned in the beginning, we think that most of the issues raised by reviewer #3 are due to misunderstanding or sometimes confusion between the spatially and spatiotemporally structuring of light. Of course, we know the computer-generated hologram (CHG) has been around for a long time. However, these have been only used in the spatial domain, not spatiotemporal domain. Similarly, we are familiar with LG beams in the spatial domain as well. The STLG wavepackets we created in this work have the LG mode profile in the spatiotemporal domain. The reviewer may have confused this with a pulse laser of spatial LG mode profile as in one of the reference the reviewer cited (Optics Express, <https://doi.org/10.1364/OE.447487>, see below for our response to comment #4).

This reviewer uses the concerns from reviewer #1 & reviewer #2 to strengthen his/her points. Now both reviewers #1 and #2 are convinced the novelty of our work. Thus the arguments raised by the reviewer #3 based on these points are no longer valid. The reviewer #3 further states **"In the response to the comment 1 of the second reviewer, it is clear from Fig. R2a and R2b the experimental implementation is quite similar expect for the CGH-based complex-amplitude modulation and iris employed in the current work, which also increase the complexity. Furthermore, the digital CGH method inducing multiple diffraction orders also causes large amount of power loss after choosing a specific diffraction order by the iris."** In other words, after reading our responses and revisions, reviewer #3 knows that the method used in this manuscript is **different** from previously reported methods with **"increased complexity"** (and there is a reason for this increased complexity, because we need to control both the amplitude and phase of the desired wavepackets, otherwise we would not be able to generate the STLG modes in this work. Why would anyone use a method with increased complexity if the same goal can be achieved with a much simpler method?).

Then if we go back and look at comment #1 from this same reviewer, where it states **"the basic principle to synthesize the spatiotemporal optical vortices based on pulse shapers (diffraction grating, cylindrical lens, 2D SLM) is exactaly the same as previous works"**. **It is not hard to see that these are actually contradictory comments.** How could he/she recognize the mechanism is different and with **"increased complexity"**, while at the same time criticize the method to be **"...exactly the same as previous works"**?

The complex-amplitude modulation technique developed in our work inevitably causes a part of energy loss since we implemented not only phase but also amplitude modulation simultaneously to sculpt intertwined structure wavepackets, otherwise we would not be able to synthesize such STLG wavepackets in this work using the previously reported methods. However, such an energy loss may be overcome in the future with the development of novel nanophotonic devices, similar to the recent developments in STOV generation. To further clarify this point, we have added the following sentence and reference in the conclusion and discussion section of the revised version:

"Moreover, while our protocol utilizes both phase and amplitude modulations, which inevitably leading to some energy loss, similar to the recent development in STOV generation [58], this drawback could be mitigated by implementing novel nanophotonic devices in the future."

58. Huo, P., Chen, W., Zhang, Z., Zhang, Y., Liu, M., Lin, P., Zhang, H., Chen, Z., Lezec, H., Zhu, W., Agrawal, A., Peng, C., Lu, Y. & Xu, T. Observation of spatiotemporal optical vortices enabled by symmetry-breaking slanted nanograting. *Nat. Commun.* **15**, 3055 (2024).

Comment 3:

3. Novelty of mode conversion between spatiotemporal LG wavepacket and an spatiotemporal HG wavepacket:

In the response file, the authors also agree that the conversion between LG beams and HG ones is a well-known technique (see [1] Phys. Rev. A 45, 335 8185-8189 (1992) [2] Nat. Commun. 10, 1865 (2019)). It is indeed not surprised the similar conversion works for spatiotemporal LG wavepacket and spatiotemporal HG wavepacket. No new physics is found in this mode conversion.

Reply:

Yes, we are honest to say that the mode conversion between LG beams and HG ones is well-known technique in the **spatial domain**. However, what we studied in the current work is the mode conversion between spatiotemporal LG and spatiotemporal HG modes. The mechanism of this conversion in the spatiotemporal domain is due to the mismatch between dispersion and diffraction. The physical mechanism is very different from its spatial counterparts. We could not agree with the reviewer's assessments that everything is obvious and can be readily achieved by simply mapping from spatial to spatiotemporal.

Comment 4:

4. Similar works of spatiotemporal wavepackets beyond spatiotemporal optical vortices:

The current work combines LG beam with spatiotemporal wavepacket, which is an extension of previous spatiotemporal optical vortices. This is similar to many previous works in this area such as spatiotemporal Bessel wavepacket, spatiotemporal vector wavepacket, spatiotemporal sculpted (tilted OAM, multiple singularities) wavepacket. All these can be regarded as extension works of the early spatiotemporal optical vortices. The current work of spatiotemporal LG wavepacket shows multi-ring topology in its spatiotemporal profile, which can be also seen in the spatiotemporal Bessel wavepacket demonstrated before (<https://www.sciencedirect.com/science/article/pii/S2095927321005120>). In particular, even for the spatiotemporal LG wavepacket, its similar idea has been proposed and studied in another previous work published on Optics Express (<https://doi.org/10.1364/OE.447487>).

Reply:

This is another example that the reviewer may have confused our spatiotemporal work with spatial works, hence underestimating the novelty and technical challenges we have to solve in

order to achieve what we reported in this work.

The reviewer's understanding that "**The current work combines LG beam with spatiotemporal wavepacket**" indicates that the reviewer has mistaken our STLG wavepackets as wavepackets with spatial LG mode profile, which is reported in the Optics Express work (<https://doi.org/10.1364/OE.447487>) cited by the reviewer. **However, these are actually completely different things.** The LG wave packet reported in this Optics Express paper has LG mode profile in the spatial domain, while our STLG modes have LG mode coupling profile in the spatiotemporal domain.

The reviewer also mentioned many other previous spatiotemporal vortice works and conclude the current work is similar to these works. We hope we have clearly explained the novelty of the current work in our responses above. It is not a straightforward extension in order to achieve what we reported in the current work.

Comment 5:

5. When carefully reading the responses to all reviewers, we found the reviewer 1's comment on the limitation of l and p values for STLG wavepackets and reviewer 2's comment on the physical difference between the generation of Gaussian-type STOV wavepackets and STLG wavepackets could help to improve the quality of the work. However, without adding new results to improve the work, especially for the limitation of l and p values, the technical advances of the revision are limited.

Reply:

We have addressed this issue in detail in the first-round response that have been recognized by the reviewer #1 and reviewer #2. To further strengthen our manuscript, we have added new experimental results for the generation of STLG wavepackets with large p and l in the supplementary section 11 to illustrate this point. However, we should understand that every technology has its limitations and one should not deny the novelty of this work simply because the achievable l and p values limited by the current devices available. Thus, we could not agree with the reviewer's statement that "the technical advances of the revision are limited."

Comment 6:

6. The authors disagree with the previous comment on the new insights into high-dimensional quantum information, photonic topology, and nonlinear optics. However, without adding supported results, how can the authors conclude the advantages of the current work of spatiotemporal LG wavepackets with controllable radial and azimuthal quantum numbers? If the basic principle and technique for synthesizing spatiotemporal LG wavepacket are similar to previous works of spatiotemporal optical vortices, one would expect advanced specific applications with the generated spatiotemporal LG wavepackets. Otherwise, the novelty and advances of the current work are not high enough to be published on Nature Communications.

Reply:

First of all, what we reported is a new type of spatiotemporal wavepackets. It is not our purpose to say that STLG has certain “advantages” over other spatiotemporal wavepackets. As we have already shown above and the reviewer also has recognized, the basic principle and technique for synthesizing STLG wavepackets are **very different** from the previous works of STOV.

We have tried our best to explain this issue in the first round and we hope the reviewer can understand that this is a general statement as a natural conclusion of what one can learn from the developments of the spatial counterparts of these spatiotemporal wavepackets. Continued studies on these topics by enthusiastic scientists can further promote the potential efficacy and versatility of STLG wavepackets in these fields, paving the way for practical applications and technological innovations. However, demonstration of these applications are far beyond the scope of a single manuscript.

In a word, the authors proposed and demonstrated spatiotemporal optical vortices with controllable radial and azimuthal indices. This work is indeed an incremental one with limited novelty compared to previous works in terms of fundamental principle of spatiotemporal wavepacket generation, complex-amplitude modulation and LG-HG mode conversion. The advances of the current work also need improvement in terms of limited values of radial and azimuthal indices (l , p) and specific practical application. Moreover, the authors do not fully and seriously respond to the comments of the reviewers and it is pity the novelty and advances of the revision do not significantly improve compared to the original version. Just providing some explanations without deep level improvement is not convincing. Therefore, this paper may be published on a more specified journal showing incremental progress of this area, but not Nature Communications aiming at publishing original and high-impact works.

Reply:

We hope that our responses above have clarified the novelty of our work. In the previous round, we responded with an 11-page response letter and revised the manuscript accordingly point-by-point. And with these responses and revisions, we are glad to see that both reviewer #1 and #2 recognize the novelty of our work now and are supportive for the publication of this manuscript in Nature Communications. Thus, we believe we have fully and seriously responded to the comments of the reviewers raised.